# Toward Sourdough Microbiome Data: A Review of Science and Patents

**DOI:** 10.3390/foods12020420

**Published:** 2023-01-16

**Authors:** Gabriel Albagli, Priscilla V. Finotelli, Tatiana Felix Ferreira, Priscilla F. F. Amaral

**Affiliations:** 1Programa de Pós-Graduação em Ciências de Alimentos (PPGCAL), Instituto de Química, Univrisdade Federal do Rio de Janeiro, da Silveira Ramos, 149, CT, Bl. A, Ilha do Fundão, Rio de Janeiro 21941-909, RJ, Brazil; 2Faculdade de Farmácia, Universidade Federal do Rio de Janeiro, Av. Carlos Chagas Filho, 373, Cidade Universitária, Rio de Janeiro21941-170, RJ, Brazil; 3Escola de Química, Universidade Federal do Rio de Janeiro, Av. Athos da Silveira Ramos, 149, CT, Bl. E, Ilha do Fundão, Rio de Janeiro21941-909, RJ, Brazil

**Keywords:** sourdough starter, microbes, patents, scientific articles, technological prospecting

## Abstract

Technological prospecting was performed on documents related to sourdough microbiota using SCOPUS, Web of Science, Google Scholar, Espacenet and Patent Inspiration databases. Scientific articles and patents were analyzed based on three different perspectives: macro (year of publication, country, and institutions), meso (categorization as different taxonomies according to the subject evaluated), and micro (in-depth analysis of the main taxonomies, gathering the documents in subcategories). The main subject addressed in patents was the starter and product preparation, while 58.8% of the scientific publications focused on sourdough starter microbiota (identification and selection of microorganisms). Most patents were granted to companies (45.9%), followed by independent inventors (26.4%) and universities (21.8%). Sourdough products are in the spotlight when the subject is the bakery market; however, a closer integration between academia and industry is needed. Such a collaboration could generate a positive impact on the sourdough market in terms of innovation, providing a bread with a better nutritional and sensory quality for all consumers. Moreover, sourdough creates a new magnitude of flavor and texture in gastronomy, providing new functional products or increasing the quality of traditional ones.

## 1. Introduction

Bread is one of the oldest fermented foods in the world. In ancient times (approximately 3100 to 332 B.C.), it was produced in a similar way to what we nowadays call sourdough bread [1]. Sourdough bread is obtained by the spontaneous fermentation or inoculation of a mixed microbial culture, named sourdough starter. Indeed, sourdough starter can be defined as a mix of water and flour fermented by yeast and bacteria—usually lactic acid bacteria—and is, traditionally, used to produce breads, panettone, pizza, buns, cakes, cookies, waffles, and pancakes [2]. However, sourdough starter also has a great potential use to reduce wheat allergens in bakery products [3], to produce beer with high and low concentration of alcohol [4,5], to obtain functional bakery products [6] or beverages such as kvass, a traditional Eastern European fermented drink [7], and to prevent fungal spoilage [8].

The differences between sourdough bread and traditional bread are mainly related to this diverse microbiota that produces a final product with different organoleptic characteristics, such as an acid flavor, a chewy and crispy texture, and a crackly crust. Moreover, the fermentation time is also very different; while traditional baker’s yeast fermentation takes about 2 h, sourdough fermentation usually takes 24 h. The fermentation conditions (temperature, pH, and moisture) are also very important to guarantee the final product quality [9,10]. The production of lactic acid by the bacterial strains also lowers the glycemic index of sourdough bread in comparison to the traditional one [11]: organic acids, such as lactic, acetic and propionic acids, could slow down the gastric emptying resulting in a lower glycemic index [12] and could transform the starch present in the flour into a resistant type, making it impossible to be digested by enzymes present in the human body [13].

Several types of flour can be used in the sourdough technological process, including the traditional wheat, but also some unconventional starch sources, such as rye, maize, quinoa, buckwheat, oat [14,15], legumes and fruits [16]. This diversity of raw materials can generate a wide variety of products with very distinct flavors. Gluten-free flours can be used to produce sourdough; and even when produced from wheat, the final product presents low gluten content due to the microbes responsible for hydrolyzing the gluten [17]. As a result, the global market for this product has been increasing in recent years. Nowadays, a gluten-free diet is the third most popular diet in the world, which consumers are adopting for different reasons such as gluten gastrointestinal sensibility, skin allergy, celiac disease, or health benefits. The increasing popularity of gluten-free food across the globe has been boosting the sourdough industry [18].

Europe is expected to dominate the global sourdough market by 2027, with Germany as the largest market in the region. France, Italy, and Spain are also important sourdough consumers. The market in the Asia-Pacific region is expected to grow at a significant rate in the next few years. The demand for baked goods free from additives and chemical residues in countries such as Japan is expected to positively impact the market growth, since sourdough products tend to be made using only the starter, flour, and water. The high availability of ingredients in this region is also another factor that helps to leverage the market [19,20,21].

In the last 5 years, fermented food increased by 149% on restaurant menus, becoming one of the most popular menu items; and in 2020, the fermented foods market was worth $9.2 billion. However, fermented foods still represent a very small portion of the food and beverage market. According to Karl De Smedt, senior communication and training manager at Puratos, during the COVID-19 pandemic, many people started baking sourdough at home, and featured their skills on social media. Thus, consumer engagement with sourdough reached a massive peak. According to research performed by Puratos, 52% of today’s consumers know sourdough: approximately 45% of them associate sourdough products with a “better taste” and nearly 30% associate sourdough with the description of being “rustic, healthier, and more natural” [22,23].

The sourdough market is highly competitive, with several domestic and multinational players “battling” for a market share. The top five players of sourdough brands are: Ernst Böcker GmbH & Co KG (Germany), IREKS GmbH (Germany), GoodMills Group GmbH (Austria), Puratos (Belgium) and Lesaffre (France) [24]. Bringing out new products that match consumer preferences and recent food consuming trends is the major strategic approach adopted by leading players. Additionally, merger, expansion, acquisition, and partnership with other companies are the common strategies to enhance company presence and boost the market [24]. Although European companies dominate leadership positions in the sourdough market, the United States also features many important players, including Alpha Baking Company, Truckee Sourdough Company, Boudin Bakery, and Gold Coast Bakeries [19,25].

Although the sourdough market is increasing and many companies have been investing in this segment, it is still an expanding market with space for new players [26]. However, the decision to invest in a certain product, process or segment is always preceded by several studies, including the local economy, market, and available technologies. Technological prospecting is a technique that can help in this process since it is based on a systematic process that examines current research, strategies and emergent technologies that can affect the future and generate greater economic and social benefits [27]. These studies aim to incorporate important information enabling a technology management approach to predict possible future technologies or conditions that can help the decision-making process [28].

One of the bottlenecks of sourdough commercialization remains the high price of sourdough products, especially for bakeries and consumers who seek high-quality products but cannot afford this on a daily basis [21]. The reason why these products are not yet economically competitive relative to traditional bread is certainly mainly related to the sourdough starter. This is also the main concern of researchers because the microbial diversity and the viability of specific microorganisms during fermentation are essential for a high-quality product. The maintenance of this diverse microbiota is time consuming because it needs to be frequently refreshed with flour and water—a procedure known as backslopping [29,30]. Some researchers have directed their study to the drying process of the starter in order to enable its use, as reviewed by Albagli et al. (2021). There are also some recent systematic reviews that gather information about microbiological, biochemical, technological, and nutritional potential, as well as sensory quality of the starter and final products [31,32]. However, a prospective study that connects scientific information and industrial knowledge related to sourdough microbiota is still missing and could drive technological investments.

In this context, this work aims to prospect the variety of technological approaches related to sourdough microbiology that are described in the scientific literature and in patents, highlighting potential technologies to support future investments and innovations. A special focus has been given to the production and use of sourdough, identifying the impact of the starter cultures on sourdough characteristics, and relating it to potential subsequent technology upgrade.

## 2. Materials and Methods

### 2.1. Pre-Prospective Step

First, a preliminary search was carried out to obtain an overview of the topic and define the keywords for the technological prospecting. This background research was developed using the SCOPUS database. Next, a more specific search was performed, based on the main keyword ‘sourdough’, resulting in a list of terminologies that was set and tested in an iterative process to validate the most representative search query in each search engine.

### 2.2. Technological Prospecting

With the search strategy defined, the documents found were correlated to short-term (five years), medium-term (five to ten years), and long-term (more than ten years) stages of technology maturity [33]. In this study, granted patents were considered short-term technologies if close to the commercial phase (likely to be in the market in the next five years). Scientific articles were considered medium and long-term technologies since they mainly report academic research; typically, they demonstrate only an initial degree of technology development, though can also sometimes report a further degree, especially when the research has been performed on a large scale and in collaboration with companies.

#### 2.2.1. Short-Term Stage: Patent Search Strategy

The Espacenet and Patent Inspiration platforms were used for the patent search. Espacenet is based on the European Patent Office (EPO) database, containing bibliographical data from around one hundred countries [34]. The Espacenet search was carried out using the following combinations of keywords: “sourdough” AND “starter” OR “microbiology”; specifically, we searched for the word “sourdough” in the title, abstract or names, and “starter” and “microbiology” in all text fields or names, resulting in the following search query: ntxt = “sourdough” AND (nftxt = “starter” OR nftxt = “microbiology”). Patent Inspiration is based on the DOCDB (EPO’s master documentation database), containing data from over 102 countries. The Patent Inspiration search was carried out using the following combination of keywords: “sourdough” AND “starter” in the title and abstract. Both searches were performed in November 2021 and the time interval chosen was from January 2010 to December 2021. Documents that were not related to the topic were disregarded after reading the abstract and claims made by the authors, and documents that were not available in English were also disregarded.

#### 2.2.2. Medium- and Long-Term Stages: Article Search Strategy

The search for scientific articles was carried out in October and November 2021 using the Scopus, Web of Science and Google Scholar databases. For the Scopus and Web of Science searches, the following combination of keywords was chosen: “sourdough” AND “starter” OR “microbiology”, in title, abstract and keywords. The combinations of keywords used in Google Scholar searches were “Sourdough Starter”, “Sourdough Microbial” and “Sourdough Microbiota”. All searches were restricted to the period from January 2010 to December 2021. Non-English documents, reviews, book chapter, conference papers, and scientific articles which were not related to the topic were disregarded.

#### 2.2.3. Data Analysis

The data analysis was divided in to three perspectives [35]:

Macro analysis: the analysis of the clearest information in the documents, such as year of publication, country of origin, authors, and institutions.

Meso analysis: the analysis of the primary information (title, abstract and keywords) to categorize the documents based on taxonomies. The taxonomies were created according to the approach of the documents, and some documents were categorized as having more than one taxonomy. The taxonomies, for both articles and patents, are presented in Table 1.

Micro analysis: the analysis of more detailed information (objective and methodology) to define, within each taxonomy, the sub-categories (or terms) by which to group them—by similarity, quantifying frequencies, and discovered patterns within the information.

## 3. Results

### 3.1. Scientific article analysis

#### 3.1.1. Macro analysis

The search for scientific articles with keywords related to sourdough starter or sourdough microbiology in Scopus, Web of Science and Scholar Google databases resulted in 791 documents. After excluding repeated scientific articles, reviews, book chapters, conference papers, and scientific articles which did not contain information related to the sourdough microbiota, a total of 515 papers were analyzed in this study. From these resulting papers, it was noted that the number of publications has grown over the last 10 years which demonstrates the increasing interest in this topic (Figure 1). Figure 1 illustrates that the scientific articles published in the last two years represent one third of the total publications in this period.

A total of fifty-five countries were involved in the publications related to sourdough microbiology. Among the top ten countries, five are located in Europe, with over 40% of total studies performed in Italy (Figure 2). Forty-four countries published less than 15 papers related to sourdough, showing that the publication profile is concentrated in only a few countries.

From the outset, it was expected that Europe would lead this research subject due to the high sourdough consumption in this region—especially Italy, which is traditionally a country associated by consumers with bakery products [36]. In 2018, its bread segment consumed 61% of the global sourdough market, followed by the pizza segment [19]. In addition to the demand for gluten-free and low glycemic index products, another issue that drives the sourdough pizza market is the increasing fast-food consumption. As sourdough pizza freezes better than standard pizza which also improves the taste considerably, the manufacturers of frozen pizza have replaced their traditional pizza with sourdough pizza. The increasing consumer receptivity to new flavors is another point that probably motivated R&D investments in the pizza industry [26].

Although European countries still publish more scientific articles related to sourdough, China’s publications have been growing since 2017. In 2019, China published the same number of articles as Italy, indicating that Asia is a potential future sourdough market.

Figure 3A,B shows the cooperation between countries in scientific research related to sourdough microbiology according to the data derived from Scopus and Web of Science, respectively. Although Italy is the major publishing country, the scientific partnership of this country is broad, collaborating with other 16 countries (Argentina, Iran, Turkey, Finland, Spain, France, Latvia, Austria, Lithuania, Belgium, Germany, China, Canada, Ireland, Nigeria, and Brazil). Meanwhile, China—the second largest current publisher and potential sourdough market in the coming years—has published in collaboration with only 8 countries (India, Italy, Canada, Nigeria, South Korea, United States, Germany, and Finland).

Of the affiliations, universities represent 75.8%, followed by research centers (18.8%) and companies (5.4%). From the top ten affiliations relating to these scientific documents, nine are universities and one is a research center, seven of them being in Europe, two in Asia, and one in Canada (Table 2). From the total of 515 scientific articles of the present study, 27.96% were performed in partnerships: most of them between universities and research centers (98), 37 between universities and companies and 9 between a university, a company, and a research center. Universities are responsible for 66.8% of the total scientific publications, followed by research centers with 5.05% and one company with 0.19% (Figure 4).

#### 3.1.2. Meso Analysis

Figure 5 shows the categorization of the scientific articles according to the meso taxonomies: microorganism, sourdough starter application, fermentation products, fermentation conditions, health, and conservation methods.

Most papers were categorized as “microorganism” (59.0%), followed by “sourdough starter application” (35.7%), “fermentation products” (23.9%), “fermentation conditions” (13.4%), “health” (7.8%), and “conservation methods” (1.9%). The category “microorganism”, which is the most commonly found topic in the papers, gathers studies related to the selection of starter cultures, isolation and identification of microbes, usage of microbes to enhance sourdough starter quality (sensory), metabolism and DNA sequencing.

These results confirm the good search strategy utilized since most papers analyzed are related to sourdough microbiota which is the main approach of this study. For this reason, the scientific papers categorized as “microorganism”, “sourdough starter application” and “fermentation products” were analyzed in a micro perspective.

#### 3.1.3. Micro Analysis

##### Microorganism taxonomy

From the studies related to the category “microorganism”, sub-categories were proposed based on the subject that they focused on (Figure 6): “identification”—involving scientific articles that identified the microbiota or developed a method of microbe identification; “microbe selection”—by which one or more microbes were selected to be used as a starter culture; “microbiota monitoring”—describing the metabolism, genome, growth rate and interactions between the sourdough microbiota; and “probiotic”— related to the usage of microbes present in the starter as probiotic. Some documents were categorized in more than one sub-category.

Most studies related to the “microorganism” taxonomy have reported the identification of the microorganisms (51.1%) and the selection of specific cultures (37.3%) (Figure 6). The identification of the species in the sourdough starter can highlight those studies that consider the impact of each individual species on the characteristics of the product—be they good or bad aspects—and also the impact of specific conservation processes, such as drying, on the microbial variety in question. In comparison to spontaneous fermentation, actively selecting the starter cultures is advantageous for controlling the final product’s sensory and functional characteristics. Belz et al. (2019) [37] used *Lactobacillus amylovorus* DSM 19280 and *W. cibaria* MG1, both previously isolated from gluten-free sourdough, in wheat bread making, and determined the minimum amount of sourdough needed to replace salt in traditional baker’s yeast wheat bread. The samples containing *L. amylovorus* showed typical cheesy notes while the usage of *W. cibaria* increased the bread volume. Both sourdough samples were described as slightly saltier than the control samples, which was explained by the production of lactic and acetic acids during fermentation, proving that the careful selection of starter cultures in sourdough starter can be used to reduce the salt added in bread making, since different microbes can produce substances with saltier flavor.

Salvucci et al. (2016) [38] isolated lactic acid bacteria (LAB) from cereals and seeds (wheat, sorghum, triticale, oat, rye, chia, and sesame) from Argentinean markets after enrichment and spontaneous fermentation. After identification via 16S rDNA, the bacteria were analyzed to measure a series of technological feature, such as: folate production, proteolytic activity, growth in presence of 6.5% of NaCl, and acidification capacity. The authors suggest that some of the bacteria could be used to improve the technological and nutritional properties of fermented foods. For example, 11 strains were able to reduce the flour extract pH below 4, which is an interesting property for sourdough to increase its rheological and flavor properties.

Aplevicz et al. (2014) [39] compared the characteristics of sourdough breads using different starter cultures, such as *L. paracasei* LP1 and LP2, *S. cerevisiae* SC1 and SC2. Breads produced with *L. paracasei* LP1 showed the lowest specific volume among the samples, while the highest specific volume was obtained by using *S. cerevisiae* SC1. The sample prepared with *L. paracasei* LP2 revealed a higher moisture content after baking than the sample prepared with *S. cerevisiae* SC2 (lowest moisture). A higher sensory score was achieved using the combination of *L. paracasei* LP2 and *S. cerevisiae* SC1, with the higher specific bread volume. This suggests the overall effect is not only due to the species chosen, because the use of specific strains can result in better quality products than others. Indeed, the mixture of strains from different species is a good strategy to obtain a product with specific characteristics.

Sourdough microbiota is usually composed by LAB and yeasts from the environment, and affected by tools and ingredients used during preparation. Wang et al. (2020) [40] isolated and identified microbes present in wheat sourdough during 15 h of fermentation using the PCR method (16S rRNA). The dominant bacteria species present in sourdough were *Fructilactobacillus Sanfranciscensis* and *Limosilactobacillus Pontis* (basonym *Lactobacillus sanfranciscensis* and *Lactobacillus pontis*, respectively), both heterofermentative LAB, widely reported in literature, and associated with the production of organic acids and flavor. The authors also described the identification of the bacterial species *Bambusa Oldhamii* and *Lolium perenne*, which had never previously been identified in sourdough. They attributed the presence of these microbes in sourdough to the microbiota of wheat flour. The dominant yeast species were *S. cerevisiae* and *K. humilis*, both frequently detected in sourdough. The prospect of sourdough production in the near future involves the use of defined starter cultures, selected to obtain a desirable characteristic: an approach that would only be possible with the identification of all the species, and certainly the most abundant ones.

Fekri et al. (2020) [41] analyzed a sourdough starter sample from Iran and identified several microbes with probiotic features (degradation of phytate, antioxidant capacity, exopolysaccharides (EPS) production, phenolic compound content and in vitro starch digestion). The authors also conducted a sensory evaluation of breads made with different combinations of these probiotic strains and observed that the breads had an overall good acceptability among the participants. Altilia and collaborators (2021) [42] investigated the effect of associating different strains of *Fructilactbacillus Sanfranciscensis* with five *Kazachstania humilis* on the growth rate and lag time values for each mixed population. The authors affirm that *F. sanfranciscensis* significantly leads the growth kinetics, affecting the ratio of bacteria/yeast cells. This data is interesting because it could predict the growth rate of two co-cultivated microbes and determine the best moment to use the starter (the moment when all microbes are most active).

It was observed that most articles related to the selection of microorganisms to be used as a starter culture use a consortium of microbes previously isolated from a sourdough starter. *Lactobacillus* and *S. cerevisiae* are the most abundant bacterial genera and yeast species, respectively, found in sourdough starters, and are also usually the most commonly isolated [3,43,44].

##### Sourdough starter application taxonomy

Approximately 35% of manuscripts were categorized as “Sourdough starter application” taxonomy, which was divided into 4 sub-categories: “Characterization” (articles with results of chemical, physical, sensory and nutritional characterization of the final products); “Impact on product” (articles considering the effect of the sourdough starter on the final product, such as shelf life); “Quality control” (publications regarding conservation methods and acrylamide reduction); and “Gluten” (publications related to gluten, such as gluten-free or reduced gluten breads). The classification of scientific articles according to the sub-categories can be seen in Figure 7: as can be seen, the majority of the scientific articles are in the “Characterization” sub-category, followed by “Impact on sourdough”, “Quality control” and “Gluten”.

For example, in the sub-category “Impact on product”, Nami et al. (2019) [45] assessed the effect of a sourdough starter composed of four *Lactobacillus* on the physicochemical properties, sensory characteristics, and shelf life of bread prepared from peal millet flour, a gluten-free flour. The authors reported that bread produced with *Levilactobacillus brevis* sourdough had the largest height and specific volume compared to all treatments and control samples. It was also observed that during storage, sourdough breads prepared with *Fructilactobacillus sanfranciscensis*, *L. paralimentarium* and *L. plantarum* starters retained the most moisture content at the end of the storage period (data not shown in the original article). It was concluded that the addition of specific sourdough starters had beneficial effects on bread quality (improving sensory scores) and shelf life.

Moreover, in the sub-category “Impact on product”, Pulkkinen et al. (2019) [46] investigated hydrolysis of vicine and convicine using endogenous β-glucosidase of faba beans in suspension combined with LAB possessing β-glucosidase activity in faba bean sourdough. Vicine and convicine can cause hemolytic anemia in people with genetic deficiency of glucose-6-phosphate dehydrogenase enzyme. The authors showed that only approximately 25% of vicine and convincine were eliminated after a 24-h incubation at pH 4.5–5.7, while in sourdough starter samples, 85% of vicine and 57% of convincine were eliminated, showing that sourdough can degrade antinutritional compounds from flours thanks to enzymes produced by the microbes. However, removal efficiency was dependent on the strain and the fermentation temperature. The importance of strain selection was thus once again shown.

Sourdough starter can also be used to eliminate mycotoxins during fermentation. Regarding the “Quality control” sub-category, Hajnal et al. (2020) [47] used whole wheat flour with 100 μg kg^−1^ of tenuazonic acid (TeA), alternariol (AOH), and alternariol monomethyl ether (AME). These compounds are mycotoxins produced by the genera *Alternaria*, which can be present during the cultivation of cereal crops depending on weather conditions. After a 48-h fermentation by sourdough starter, the content of TeA, AOH and AME decreased by 60.3%, 41.5%, and 24.1%, respectively.

Montemurro et al. (2019) [48] used germination and selected sourdough starter to improve the functional and nutritional quality of breads produced with wheat, barley, chickpea, lentil, and quinoa sprouted flour. The authors selected different *Lactobacillus* species (*Lactobacillus rossiae* LB5, *Lactobacillus plantarum* 1A7 and *Fructilactobacillus sanfranciscensis* DE9) as starter culture and observed an increase in total phenols and antioxidant activity in comparison to the raw grains. In addition, the release of more peptides, free amino acids, and soluble fibers also contributed to enhancing the nutritional quality of the final product. For all cereals (wheat and barley), legumes (chickpea and lentil), and pseudocereals (quinoa), germination and fermentation by selected LAB increased protein digestibility and antioxidant activity and reduced starch availability. The authors were able to relate sensory attributes of the breads to the different types of flour: the overall taste, considered as a global index of the palatability, was higher in sprouted wheat, barley, and lentil compared to the corresponding raw flours, and was lower in chickpea and quinoa. The most appreciated bread was the wheat, while the lowest score corresponded to chickpea. Industries are seeking products with those features because consumers are demanding foods with higher nutritional qualities, as this study has demonstrated.

The majority of gluten related studies involve the use of sourdough starter to produce gluten-free bread with different grains and legumes, such as sorghum, quinoa, corn, millet and acorn [49,50,51]. There are also some articles that investigate the partial degradation of gluten during sourdough fermentation, leading to a bread that would be suitable for celiac patients [52,53,54]. One of these studies demonstrated that some specific LAB strains are able to perform the proteolytic degradation of wheat proteins albumin/globulin and gliadin, considered wheat allergens [52]. The use of these strains for sourdough production is highly effective for obtaining reduced gluten bread. However, the complete elimination of the allergenic proteins is not possible with fermentation alone because the proteolytic activity of the LAB strains is not high enough. In fact, some argue that gluten is an important component of bread, even if only in small amounts. Cakir et al. (2021) [43] produced sourdough using hull-less barley or wheat flour in different proportions, using a specific combination of several LAB with yeast to develop the bread features. Though this combination increased the quality of the barley flour breads, the effect on control breads using sourdough produced from wheat flour alone was higher still, showing gluten to be an important factor.

##### Fermentation products taxonomy

The main fermentation products generated by sourdough microbiota are mostly related to sensory attributes, such as volatile and phenolics compounds, and organics acids [44,55,56,57,58]; however, some researchers have also found phytase, γ-aminobutyric acid (GABA), antioxidant and anti-inflammatory compounds, thus associating sourdough consumption with potential health benefits [41,56,59,60,61,62].

The sub-categories proposed for this taxonomy are the following: “Bioactive”, that gathers studies involving compounds with an effect on a living organism, such as antioxidants, and GABA; “Volatile compounds”, with papers related to volatile compounds present in the starter or final product; “Carbohydrates”, which considers papers related to the production of EPS, fructooligosaccharides, dextran and oligosaccharides; “Antibiotics”, with studies focusing on the production of compounds related to growth inhibition of deteriorating fungi; “Organic acids”, that refers to studies mainly reporting on the production of organic acids; “Sensory”, considering the impact of organic acids and other compounds (e.g.: volatiles, amino acids, alcohols, and organic acids) on sensory evaluation of sourdough breads; and “Metabolism”, which comprise scientific articles regarding the metabolism of the microbes. The classification of the scientific articles according to these sub-categories is presented in Figure 8. The “Bioactive” sub-category leads the taxonomy with 38 scientific articles, followed closely by “volatile compounds” with 32 papers. The “Bioactive” sub-category leading these studies is interesting from a marketing perspective since it shows that different sourdough starters can be used to create new functional food products.

Organic acids, volatile and phenolics compounds detected in sourdough are the main metabolite products studied in the documents related to “fermentation products”. The predominant organic acids are acetic and lactic, both produced during fermentation by hetero- and homofermentative *Lactobacilli* present in the sourdough microbiota and deeply connected to sensory aspects, quality, and safety of fermented food in general [55]. Organic acids are also important because of their anti-spoilage activity, and thus, by acidification, enhancing the shelf life of products [63].

Palla et al. (2020) [56] characterized 139 yeasts isolated from cereal-based fermented foods for their phytase and antioxidant activities during sourdough fermentation on different wholegrain flour (a conventional, red-grained wheat variety; a yellow-grained wheat variety rich in carotenoids; a blue-grained wheat variety rich in anthocyanins; a winter emmer variety; and a hull-less spring barley variety). Higher phytase and antioxidant activity could be achieved on certain flour types by selected *S. cerevisiae* strains, showing the importance of selecting flours and yeasts to produce a functional sourdough bread.

Two articles studied the production of GABA [62,64], a neurotransmitter found in the central nervous system and in peripheral tissues. This molecule acts as an inhibitor in human neurons, having a wide variety of functions on human body (relaxation, hypotension, antidepressant, chronic kidney disease action, immunity increase, tumor suppressor, etc.) [65]. Bhanwar et al. (2013) [62] used *Lactococcus lactis* isolated from yam pickle after screening for GABA producing bacteria. The selected strain was able to produce 226.22 mg of GABA per 100g of bread, demonstrating that *Lactococcus lactis* would be a good species as a starter in the production of GABA-containing functional fermented foods, such as sourdough breads. Cataldo et al. (2020) [64] screened 17 bacteria isolated from quinoa or Andean amaranth sourdough to produce GABA and *Levilactobacillus brevis* CRL 2013 produced the highest GABA levels (258 mM) after 72 h of fermentation. This strain of *Levilactobacillus brevis* could also be used as a starter culture for a sourdough with functional activity.

Exopolysaccharides (EPS) was another fermentation product found in research articles in the pool of publications categorized as “Fermentation products” taxonomy. EPS are extracellular polymers produced by bacteria, and in sourdough, it is produced by LAB. Two classes of EPS from LAB can be produced: homopolysaccharides (HoPS) (composed of only one type of monosaccharide) and hereopolysaccharides (HePS) (composed of multiple types of monosaccharides). EPS can affect the dough rheology and bread texture, and also have prebiotic properties [66]. Bockwoldt and collaborators (2020) [67] studied the production of β-glucan, a type of EPS, by *Levilactobacillus brevis* TMW 1.2112 and *P. claussenii* TMW 2.340 in wheat and rye sourdough. *Levilactobacillus brevis* TMW 1.2112 was a suitable starter culture for stable β-glucan formation in wheat and rye doughs, showing a higher viscosity than the control doughs and the dough prepared with *P. claussenii* TMW 2.340.

While the “Bioactive” sub-category gathers more papers overall, the “Volatile compounds” sub-category included the highest number of articles published in a single year (10 articles in 2021). The main chemical classes found in those studies are alcohol and esters followed by organic acids and aldehydes [40,68,69,70].

### 3.2. Patent Analysis

#### 3.2.1. Macro Analysis

The search on the Espacenet and Patent Inspiration databases resulted in 92 and 53 documents, respectively, related to sourdough microbiology. The repeated ones as well as patents containing non-relevant information about sourdough starter were disregarded, resulting in a total of 87 documents.

In general, the number of granted patents has increased over the last years, as also observed in the analysis of scientific articles, reaching 16 documents in 2020 (Figure 9). Other publication peaks can also be observed (in 2017, 2015 and 2014).

The documents related to sourdough microbiology were mostly registered in South Korea (35 documents). Unlike the scientific articles, the Asian continent leads this category since around half of total patents are registered in South Korea and China (Figure 10).

Among the top ten patent holders, four of them are industries and none of them is in Italy (Table 3). Indeed, Figure 11 shows that 45.98% of these patents were granted to companies and only 21.84% to universities. Only three partnerships between universities and companies and one partnership between a research center and a company were observed in the documents analyzed. The major patent holders are “Paris Croissant”, which is a bakery in South Korea and “Fed Gosudarstvennoe Avtonomnoe Obrazovatelnoe Uchrezhdenie Vysshego Obrazovaniia”, a university in Russia. The other companies registered 3 or less patents, indicating a competitive market.

The profile observed in the distribution of patent holders is unusual, since 26.44% are inventors and not a company or a research group. It can be assumed that this is due to the participation of bakers or small regional producers.

#### 3.2.2. Meso Analysis

The patents were categorized according to the proposed taxonomies, resulting in 82.8% relating to “sourdough starter or product preparation”, 21.8% relating to “microorganism” and 3.4% relating to other topics (“others”). The sum of percentages exceeds 100% since some documents were categorized as belonging to more than one taxonomy.

One of the patents that resulted from a collaboration between a university and a company manufacturing sourdough breads describes the usage of flour made from crickets in their process (KR102272552B1) [71] (present in “sourdough starter or product preparation” taxonomy), a subject discussed in two different scientific articles [55,72]. Nissen et al. (2020) [55] refers to the use of cricket flour for protein fortification in gluten-free sourdough. The authors affirm that the resulting antioxidant activity was improved, and that the use of cricket powder conferred a typical aroma to the bread. Galli et al. (2020) [72] described the technological feature of LAB isolated from fermented cricket powder as a starter culture, proving that cricket flour could be used in sourdough bread making because the bacteria isolated showed a good aminopeptidase activity, acidification, and growth rate.

The two following patents are present in the “microorganism” taxonomy: the patent AU2020206861A1 was granted to the company Puratos NV in 2021 and describes the usage of different *Clostridium saccharobutylicum* strains as a starter culture in bakery and patisserie products since it was observed that the physical properties and sensory characteristics were improved; the patent granted to Shmakova Svetlana Valerevna (RU2752999C1) [73] reports the usage of different *Lactobacillus* species and *Saccharomyces pastorianus* to develop a gluten-free starter to be used in the production of gluten-free muffins with different types of flour.

Two other patents describe the development of an incubator for sourdough starter (present in the “Others” taxonomy). The patent CA3027405A1 was granted to Alexandre and Joshua in 2020 [74] and describes a machine that maintains the culture by performing the backslopping process automatically after adding water and flour and controlling the temperature of fermentation. The document KR20170089100A granted to Yeun and Tae in 2017 [75] describes a device capable of producing a starter by adding the ingredients (raw materials and microorganisms), mixing, and controlling the temperature of fermentation. No scientific articles have approached this subject; however, this was expected since both patents describe inventions that need to be protected. It is therefore a potential field for research in the sourdough market since it facilitates the usage of sourdough starter.

## 4. Discussion

Sourdough bread is traditional in Europe, but the Asian market has been gaining strength in the last few years and could even take the lead in the market: indeed, South Korea was the largest sourdough patent holder in the last decade, followed by China. China has also increased their sourdough studies in the last 5 years, publishing as many scientific articles as Italy in 2019. However, the published scientific articles about sourdough in the last decade were mainly focused on identification and selection of microorganisms to be used as sourdough starter. Although sourdough fermentation is commonly carried out through backslopping—a protocol characterized by daily propagation—which results in the dominance of the best adapted strain, the present systematic review shows that scientists rely instead on the use of specific starter cultures to obtain well-defined characteristics in sourdough. Backslopping represents a cheap and reliable preservation method, reducing the risk of fermentation failure, which can result in spoilage and the survival of pathogens, creating unexpected health risks in food [76]. This is important for small producers, such as bakeries, but for large-scale production, in industries that follow sanitary rules, the use of selected starter cultures can be a perfect strategy.

There are also several published studies reporting on the characterization of sourdough products and the impact of the starter on these products, as well as on the fermentation products and their relationship with those characteristics (quality, sensory, nutritional, shelf life, among others). All this knowledge is extremely important for the technology since it informs the selection of the microorganisms present in the starter, according to which sourdough product and its characteristics are needed. On the other hand, the sourdough patents over the last decade have not only considered the identification and selection of microorganisms as their main subject, but also the preparation of the sourdough starter and final products. The preparation of the sourdough starter and/or products were described in 82.8% of the total number of patents granted over the last 11 years, which demonstrates that there is an interest in improving and protecting the production processes of sourdough products. Of these, 43% feature the use of non-traditional ingredients, such as algae, citron and shiitake mushroom, to produce a sourdough starter. The other 57% of the patents in question variously describe optimization, development of gluten-free or functional products, and sensory enhancement.

Regarding the sourdough patent holders, 45.98% of them are companies and 26.44% are independent inventors (the independent inventors could be small producers that make sourdough goods for small communities). Among the companies, none has a significant number of granted patents: this shows the large number of small players and individual entrepreneurs in the global sourdough market. The company Puratos was the only company present in the top 10 patent holders that is also listed as one of the greatest players in the market [24]. Universities alone also have a productive participation in sourdough patents (21.84%), but their collaboration with companies was not meaningful (3.45%).

A low collaboration between universities and companies was also observed in scientific articles demonstrating a lack of partnership between these two types of organizations. Around 66.8% of the total number of scientific articles on sourdough in the last 11 years were published by universities alone, 19% by universities in collaboration with research centers, and only 7.18% by collaborations between universities and companies. These results indicate that many of these new discoveries about sourdough might be withheld in research institutions and may never be converted to an innovation in the sourdough market.

According to Nielson-Stowell (2022) [77], the fermentation process is motivating scientists to interact more with gastronomers. The director of the fermentation laboratory at Noma (a two-star Michelin restaurant in Copenhagen), Jason White, says that “there is going to be a huge exchange in the innovation in flavor coming from gastronomy and innovation from a high-level biotechnology coming from academia, thus, making it possible to create a new infrastructure and community of people that have the same goal”. The prospect is thus that gastronomy will increasingly use fermentation to enhance flavor and change texture, in a way that cannot be achieved by other common culinary processes, such as baking and freezing. In this sense, scientists will have to be aware of these needs so they can develop innovative fermentation technologies that meet these demands. The selection of new sourdough starters (performed by scientists) that results in products with specific characteristics (demanded by gastronomers) is one of the examples of this interaction.

The researchers in academia and strategists in industries need to communicate, since most of the patents report the development of a starter or products while the companies focus on the market. Within academia, there is experience of a wide diversity of microorganisms and knowledge related to the impact of these microorganisms’ metabolism on sourdough products that could help develop specific types of texture or flavor in the final product. Thus, with this interaction, sourdough can indeed deliver a new magnitude of flavor and texture in gastronomy, providing new functional products that can be obtained through microbiota selection, increasing sourdough consumption and, consequently, profit for the bakery industry.

## Figures and Tables

**Figure 1 foods-12-00420-f001:**
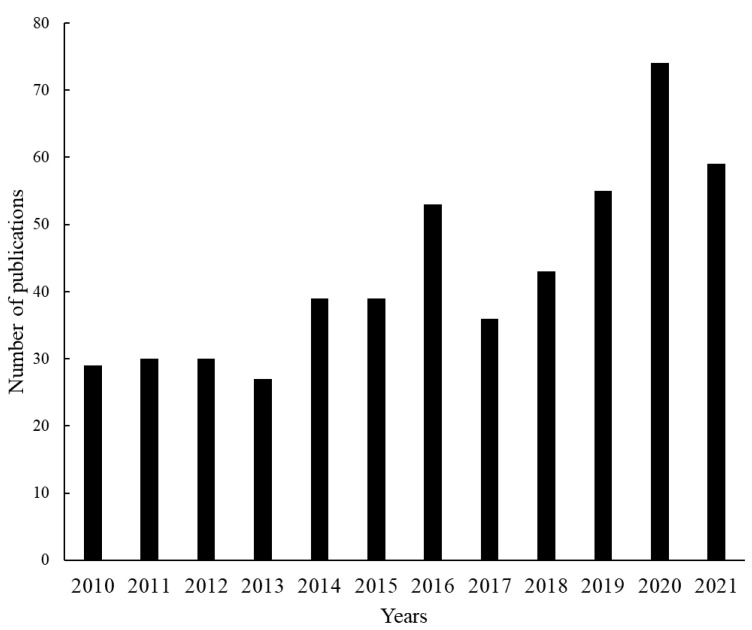
Total publications over the years related to sourdough starter/microbiology according to Scopus, Web of Science and Google Scholar databases.

**Figure 2 foods-12-00420-f002:**
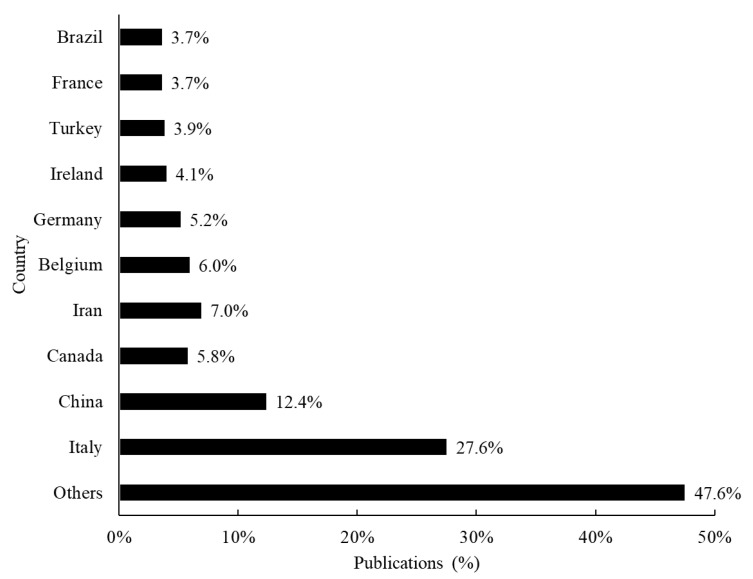
Scientific articles related to sourdough starter/microbiology per country over the last 10 years according to Scopus, Web of Science and Google Scholar databases.

**Figure 3 foods-12-00420-f003:**
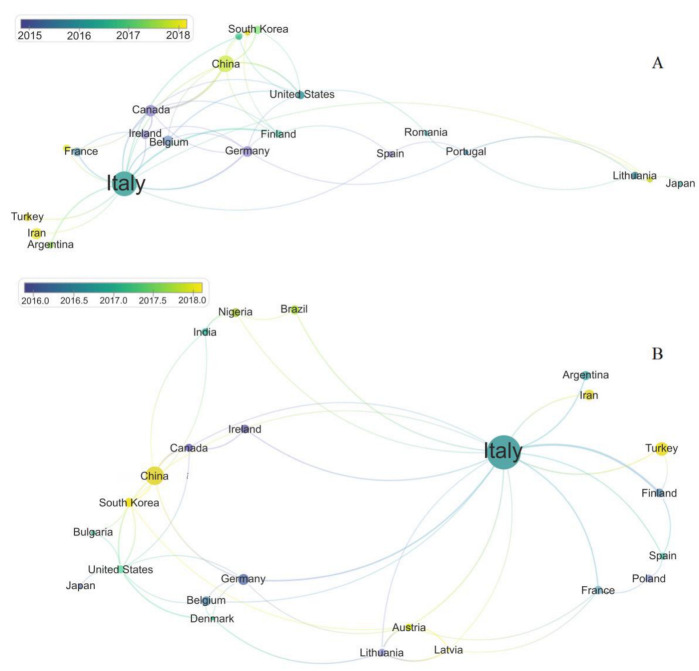
Partnerships between countries in the scientific articles reporting on sourdough starter/microbiology in Scopus (**A**) and Web of Science (**B**) databases (VOSviewer software). The size of the nodes (name of the countries) indicates the number of publications, node colors indicate the average publication year of the articles, and the lines indicate the partnership: the thicker the line, the more publications the countries have in conjoint.

**Figure 4 foods-12-00420-f004:**
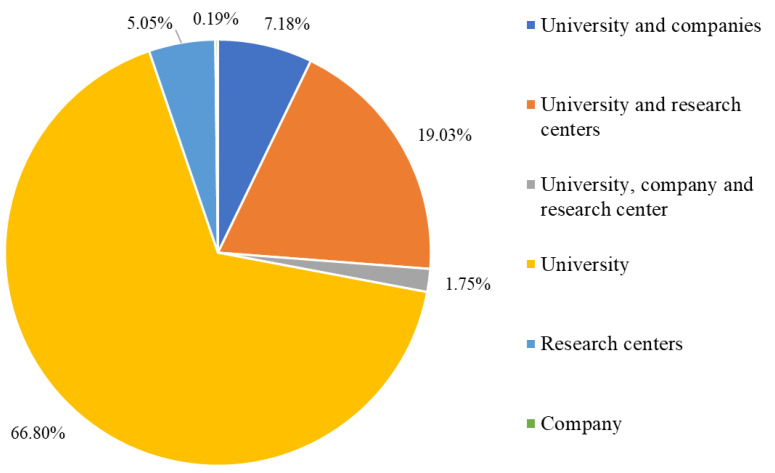
Percentage of the types of affiliation and partnership involved in the sourdough starter/microbiology scientific articles over the last 10 years according to Scopus, Web of Science and Google Scholar databases.

**Figure 5 foods-12-00420-f005:**
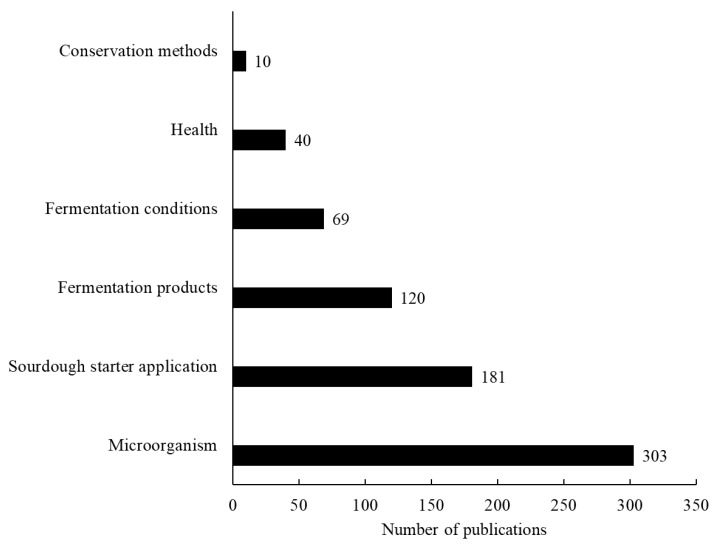
Categorization of the scientific articles related to sourdough starter/microbiology according to the taxonomies proposed in the meso analysis.

**Figure 6 foods-12-00420-f006:**
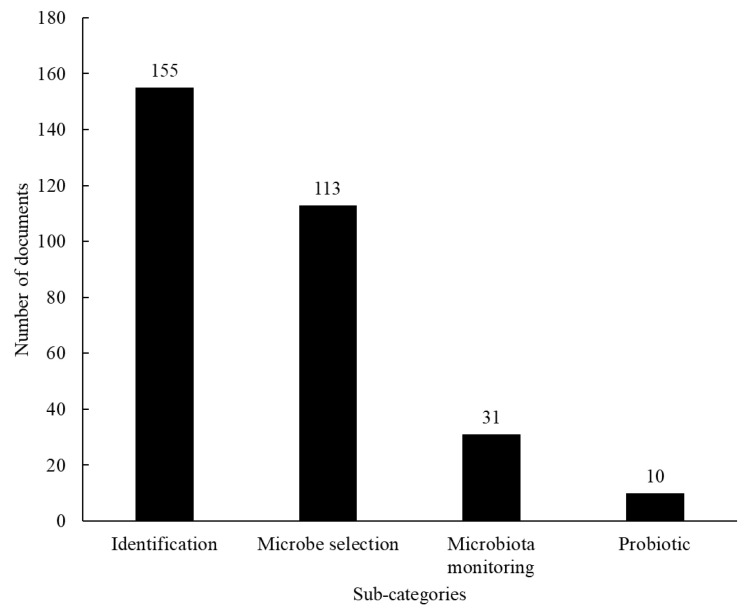
Classification of the scientific articles categorized as “microorganism” taxonomy according to the sub-categories proposed in the micro analysis.

**Figure 7 foods-12-00420-f007:**
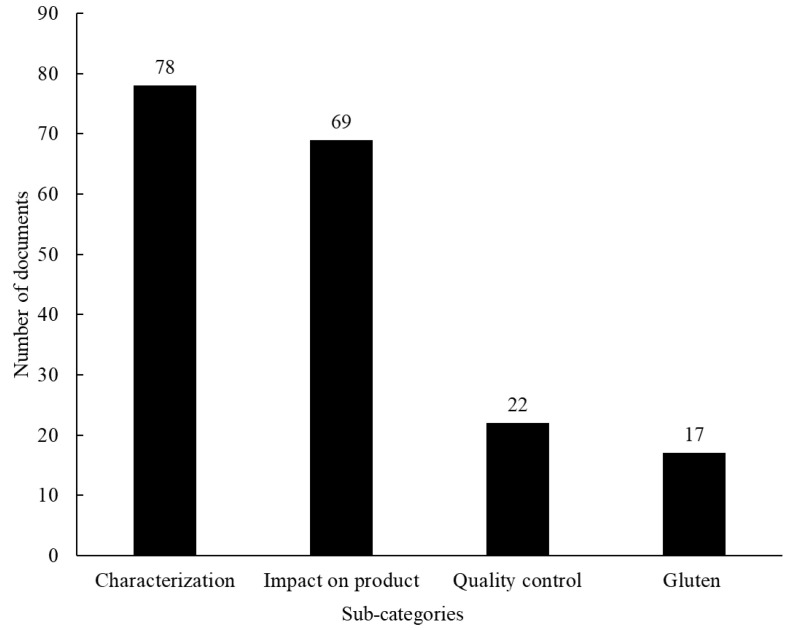
Classification of the scientific articles categorized as “sourdough starter application” taxonomy according to the sub-categories proposed in the micro analysis.

**Figure 8 foods-12-00420-f008:**
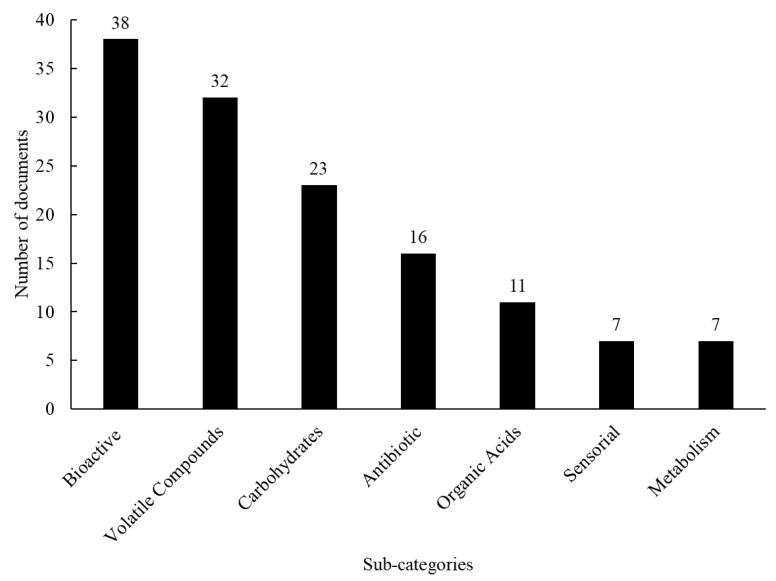
Classification of the scientific articles categorized as “fermentation products” taxonomy according to the sub-categories proposed in the micro analysis.

**Figure 9 foods-12-00420-f009:**
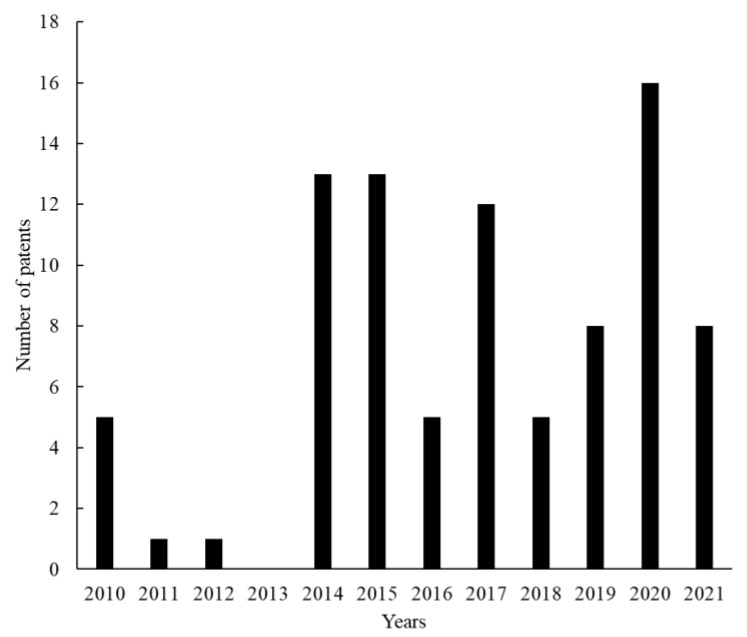
Total granted patents related to sourdough microbiology over the last 10 years according to Espacenet and Patent Inspiration databases.

**Figure 10 foods-12-00420-f010:**
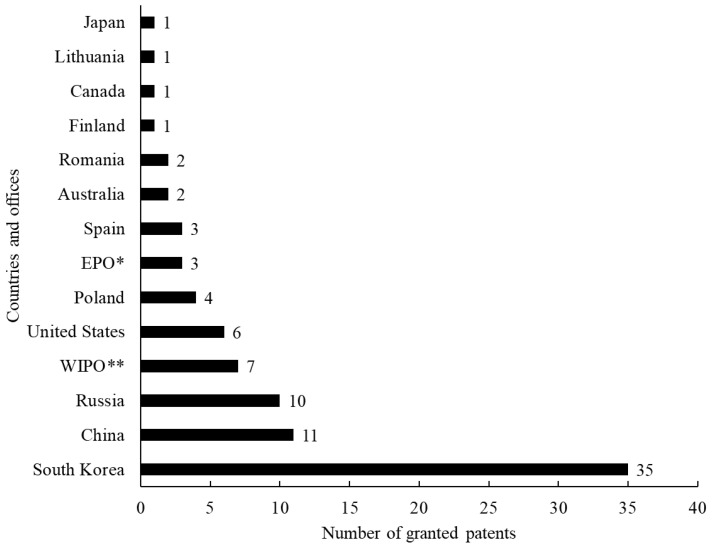
Number of patents granted in different countries according to Espacenet and Patent Inspiration databases. *EPO—European Patent Office; **WIPO—World Intellectual Property Organization.

**Figure 11 foods-12-00420-f011:**
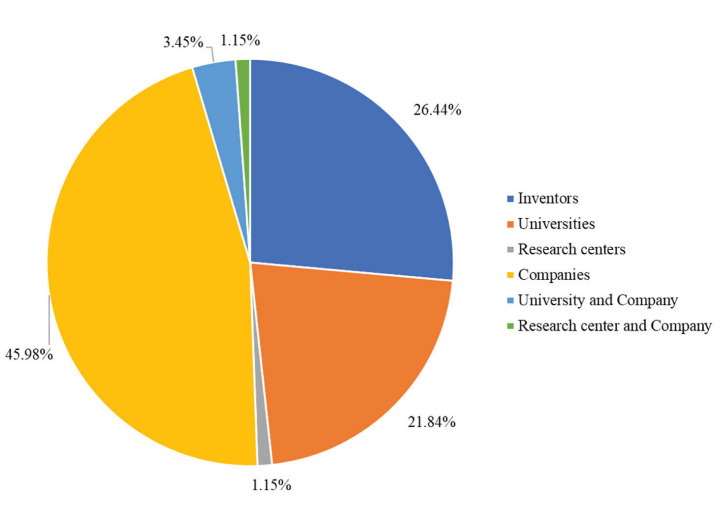
Percentage of the types of affiliation and partnerships involved in the sourdough patents over the last 10 years according to Espacenet and Patent Inspiration databases.

**Table 1 foods-12-00420-t001:** Taxonomies created to categorize the scientific articles and patents.

Type of Document	Taxonomy	Definition
Scientific Articles	Sourdough starter application	Publications in which microbiota of sourdough starter is related to bakery product characteristics (quality and sensory)
	Microorganism	Publications related to the selection of starter cultures, isolation and identification of microbes, usage of microbes to enhance sourdough starter quality (sensory), metabolism and DNA sequencing
	Fermentation products	Publications regarding the metabolites produced by the sourdough starter microbiota
	Fermentation conditions	Publications in which fermentation conditions are the main subject
	Health	Publications relating sourdough starter microbiota to health improvement
	Conservation method	Publications that describe different drying methods to increase the shelf life of the starter
Patents	Microorganism	Documents related to the selection of microbes to be used in the starter
	Sourdough starter or product preparation	Documents that describe the preparation of the starter and/or a final product, such as breads, panettones, and pizzas
	Others	Documents with no relation to any other taxonomies

**Table 2 foods-12-00420-t002:** Top 10 affiliations of sourdough starter/microbiology scientific articles over the last 10 years according to Scopus, Web of Science and Google Scholar databases.

Affiliation	Number of Publications	Country
Università degli Studi di Bari	53	Italy
University of Alberta	30	Canada
University College Cork	21	Ireland
Vrije Universiteit Brussel	21	Belgium
Jiangnan University	20	China
Technical University of Munich	15	Germany
Consiglio Nazionale delle Ricerche	14	Italy
Free University of Bozen-Bolzano	14	Italy
Università degli Studi di Milano	14	Italy
Gorgan University of Agricultural Sciences and Natural Resources	14	Iran
Others	305	

**Table 3 foods-12-00420-t003:** Top 10 sourdough patent holders according to Espacenet and Patent Inspiration databases.

Patent holder	Number of patents	Country
Fed Gosudarstvennoe Avtonomnoe Obrazovatelnoe Uchrezhdenie Vysshego Obrazovaniia	6	Russia
Paris Croissant CO LTD	6	South Korea
Puratos NV	3	Belgium
Shmakova Svetlana Valerevna	3	Russia
Zhejiang University	3	China
DIOSNA Dierks & Söhne GmbH	2	Germany
Jiangnan University	2	China
Tianjin University of Science and Technology	2	China
Europastry S A	2	Spain
Fall In Bread CO LTD	2	Korea
Others	56	

## Data Availability

Not applicable.

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
