# Peer review of "Toward Sourdough Microbiome Data: A Review of Science and Patents"

_foods, 2023, doi:10.3390/foods12020420_

Round 1

Reviewer 1 Report

Review of the manuscript

Manuscript ID: foods-2111449

Title: Toward Sourdough Microbiome Data: a review of science and patents

The manuscript presents an excellent work of an in-depth analysis of the literature data and patents published in the last decade about the sourdough microbiota. The manuscript has a good structure, and I think it is easy for the reader to follow the topic.

In my opinion this manuscript is a Review, not an Article. Maybe the authors should reconsider the manuscript type.

Reviewer 2 Report

The taxonomy focused on starter culture and fermentation in the Table 1, taxonomies created to categorize scientific articles and patents, but there is no data for material type (wheat, rye, maize, 54 quinoa, buckwheat, oat,...).There is also no data for the articles published in safety issues, such as the health risk assessment of process-related contaminants in sourdough.

Reviewer 3 Report

This review paper is aimed at providing a technological prospecting of sourdough industry. Authors have spent a lot of effort in reviewing scientific publications and patents from various search engines. However, in my opinion, the study is not comprehensive. It is believed that this review article would be able to fetch high number of citations in the new future if the content is focused on state of the arts of sourdough microbiome technology and its future prospect. The status of the manuscript lacks focus, and the discussion brought forward is not in depth. Authors are advised to critically analyze the data presented in the literature and provide more significant findings and its impacts on the advancement of the sourdough industry. Below listed some flaws detected in the manuscript:

Page 1, Line 37: Do you mean sourdough starter??

Page 2, line 51: to resistant starch……

Page 7, Figure 3: Authors are advice to label properly the name of the countries reported in Figure 3. . Besides, it is recommended to combine the analysis acquired from Scopus and Web of Science to better reflect the scenario.

Page 10, Line 276: The remark is invalid. Salt added to bread dough serves other functions as well, in addition to making the final bread taste salty.

Page 13, Line 373: Change “indicated” to “suitable”.

Page 14, Line 422: 72 h or ??

Page 14, Line 424: Define EPS.

Page 17, Line 517: Change “culinary” to “technology”.

Page 17, Line 520: ……not only the identification …..

Page 18, Line 547-550: The statement needs further elaboration.

Page 18, Line 551-557: The idea flow of this last paragraph needs restructuring.

Round 2

Reviewer 3 Report

The manuscript has been well revised. Appreciate efforts given by the authors. Thus, I recommend acceptance for publication.